# Biofilm and Cancer: Interactions and Future Directions for Cancer Therapy

**DOI:** 10.3390/ijms241612836

**Published:** 2023-08-16

**Authors:** Euna Choi, Ben Murray, Sunga Choi

**Affiliations:** 1Department of Biology, Union University, Jackson, TN 38305, USA; echoi@uu.edu (E.C.); benjman13@gmail.com (B.M.); 2Department of Bioinformatics and Biosystems, Seongnam Campus of Korea Polytechnics, Seongnam-si 13122, Republic of Korea

**Keywords:** extracellular polymeric substance (EPS), tumor microenvironment (TME), tumor microbiome (TM), tumor-associated macrophages (TAM), colorectal cancer (CRC), *Fusobacterium nucleatum* (*Fn*), *Helicobacter pylori* (*Hp*)

## Abstract

There is a growing body of evidence supporting the significant role of bacterial biofilms in the pathogenesis of various human diseases, including cancer. Biofilms are polymicrobial communities enclosed within an extracellular matrix composed of polysaccharides, proteins, extracellular DNA, and lipids. This complex matrix provides protection against antibiotics and host immune responses, enabling the microorganisms to establish persistent infections. Moreover, biofilms induce anti-inflammatory responses and metabolic changes in the host, further facilitating their survival. Many of these changes are comparable to those observed in cancer cells. This review will cover recent research on the role of bacterial biofilms in carcinogenesis, especially in colorectal (CRC) and gastric cancers, emphasizing the shared physical and chemical characteristics of biofilms and cancer. This review will also discuss the interactions between bacteria and the tumor microenvironment, which can facilitate oncogene expression and cancer progression. This information will provide insight into developing new therapies to identify and treat biofilm-associated cancers, such as utilizing bacteria as delivery vectors, using bacteria to upregulate immune function, or more selectively targeting biofilms and cancer for their shared traits.

## 1. Introduction

Biofilms are estimated to be involved in ~80% of all human infections as well as various types of human pathologies, including cancer [1]. Biofilms are communities, often multispecies communities, of microorganisms enclosed within a self-constructed matrix of extracellular polymeric substances, such as polysaccharides, proteins, extracellular DNA, and lipids [2]. This matrix can either adhere to a surface, such as skin, wound tissue, or indwelling medical devices, or become suspended within a secondary matrix of host-derived pus or mucus [3]. Over time, biofilms mature into a mushroom-shaped structure, and at the end of their development can release microorganisms to colonize new surfaces and serve as reservoirs of infection.

Biofilms contain various microorganisms, even multicellular organisms. The physiological states of these organisms are heavily contingent upon their location both within the body and within the biofilm. The limited diffusion of chemicals within a biofilm creates gradients for O_2_, pH, and nutrients, with the center zones often being hypoxic and nutrient poor. Microorganisms within the biofilm may, therefore, be actively dividing, non-dividing, metabolically active, or inactive depending on their conditions within the biofilm microenvironment [4]. Organisms in a biofilm can also respond to environmental stimuli via quorum sensing, using small signaling molecules that can up- or down-regulate stress tolerance, metabolic, or virulence genes in the colony [5].

Emerging research suggests a link between biofilms and various types of cancer, even beyond the conventionally accepted links to colorectal cancer (CRC) and gastric cancer. Malignant cells in vivo are supported by a tumor microenvironment (TME) that consists of various immune cells, fibroblasts, vascular endothelial cells, and adipocytes that communicate via chemokines, cytokines, growth factors, exosomes, proteoglycans, and glycoproteins [6]. The TME thus contributes to the cellular and genetic heterogeneity encapsulating tumors, immune cells, and bacteria, creating an abnormal environment that remodels the extracellular matrix (ECM) [7]. The altered ECM triggers changes in gene expression, leading to epithelial-to-mesenchymal transition (EMT), a hallmark of tumor cell migration and invasion. Moreover, the TME modulates immune cell functions, shifting them from pro-inflammatory to anti-inflammatory states. Regulatory T cells suppress immune responses, hindering the effector T cells’ ability to attack tumors. Myeloid-derived suppressor cells and tumor-associated macrophages also inhibit T cells and natural killer cells, further suppressing immune responses. In addition, tumor cells and other cells in the TME release immunosuppressive cytokines.

As a multicellular aggregation, cancer cells must be able to efficiently distribute nutrients, requiring metabolic changes collectively known as the Warburg effect. This metabolic transition in cancer cells allows sufficient ATP generation under hypoxic conditions and provides an elevated level of glutamine, a viral substrate for cell growth. These changes can be accomplished by increased expression of transporters for the substrate and enzymes involved in its metabolism [8]. Similarly, biofilms are biological structures formed through cooperative interactions among microorganisms to ensure their survival. Within biofilms, microorganisms have developed specialized adaptive mechanisms to thrive in extreme environments, akin to the Warburg effect observed in cancer cells. These microorganisms efficiently metabolize glucose to derive energy and maintain the stability of the biofilm, even in oxygen-limited conditions [9]. The commonality between biofilms and cancer cells lies in their unique metabolic adaptations for nutrient distribution, supporting their survival and growth. Cancer cells sustain rapid proliferation by employing the Warburg effect to acquire high levels of substrates such as glutamine [10]. Understanding the mechanisms behind these metabolic adaptations is a crucial challenge in cancer therapy and biofilm control.

Biofilms can promote cancer through various mechanisms (Table 1), as follows: (1) Biofilms can trigger inflammation that often fails to clear biofilm-associated pathogens, and persistent inflammation can cause DNA damage and promote the growth of cancer cells [1]. (2) Biofilms can modulate and limit the efficacy of the host immune response [11], creating an environment that supports cancer development [12]. (3) Some bacteria in biofilms can produce toxins functioning as carcinogens, which increase the risk of cancer [13]. (4) Bacteria in biofilms can alter host metabolism. (5) New evidence increasingly suggests that bacteria within the TME, referred to as the tumor microbiome (TM), are involved in cancer progression [14]. Biofilms can alter the microenvironment surrounding cancer cells. The altered ECM protects cancer cells and facilitates their invasion and metastasis. Considering the prevalence of biofilm formation in the human body and its detrimental impact on infections and cancer progression, there is an increasing emphasis on gaining a deeper understanding of bacteria behavior in cancer. Elucidating the mechanisms of biofilms’ contribution to cancer will allow us to improve the diagnosis and treatment of biofilm-associated cancers [15] and will also lead to new and innovative treatment options. This review will primarily discuss the relationship of biofilms with cancers such as CRC and gastric cancer. The arising association between biofilms and other types of cancer will also be discussed.

## 2. Biofilm and Colorectal Cancer (CRC)

Colorectal cancer (CRC) is one of the leading causes of cancer-related death worldwide [24]. Among various contributing factors, dysbiosis, an imbalance in the microbiota, appears to play a key role in its pathogenesis [25]. Research has found that patients with CRC have distinct microbiota compositions compared to healthy controls. CRC tissue contains elevated levels of an invasive anaerobe, *Fusobacterium nucleatum* (*Fn*), compared to healthy tissue, which is linked to lymph node metastasis [26,27]. In addition to dysbiosis, CRC is also associated with a compromised mucosal barrier [28]. The colon is lined with a dense layer of mucus that separates the microbiota from the human colonic epithelial cells. This mucus layer prevents direct contact between the microorganisms and the cells, but these mucosal barriers can be compromised when bacteria form biofilms, leading to chronic mucosal inflammation that eventually damages colonic epithelial cells (Figure 1A,B). Disruption of the colonic epithelial cells can result in the loss of E-cadherin, a cell–cell junction responsible for connecting epithelial cells, from the apical zonula adherens [29]. This condition is also accompanied by elevated levels of pro-inflammatory cytokines such as IL-6, increasing the likelihood of CRC development, particularly in individuals with obesity, diabetes mellitus, and smoking habits [30].

Damage to the mucus barrier allows microorganisms direct contact with colonic epithelial cells and can alter the mechanisms by which microorganisms metabolize, dramatically affecting nearby host cells. Heightened levels of acetylated polyamine metabolites such as N 1-acetylspermidine and N 1-acetylspermine have been found in biofilm-concentrated human colon cancer, driven by altered levels of bacterial acetyltransferases [31]. Polyamines are naturally occurring polycations in eukaryotes that can stabilize or distort the conformation of negatively charged macromolecules in cells, such as DNA, RNA, proteins, and acidic lipids containing phosphate groups. They play a critical role in cell growth and differentiation [32]. The polyamine pathway is a downstream target for many oncogenes, such as MYC transcription factor, p53, RAS, MEK, AKT, and mTOR [32]. Dysregulation of polyamine metabolism is frequently observed in other cancer cells, such as lung, breast, prostate, and gastric cancers [32]. This indicates that bacteria in biofilms manipulate the host polyamine metabolism, ultimately accelerating cell growth and tumor formation.

Deoxycholic acid (DCA) is another important metabolic modulator found in human colon biopsies and mouse models, mediating CRC progression [33]. DCA promotes DNA damage and oxidative stress, enhancing proliferation of tumor cells, and suppressing apoptosis, functioning as a naturally produced carcinogen [34]. It is one of the secondary bile acids produced by the host but modified by intestinal bacteria. Biofilms can provide an optimal microenvironment where bile acids are changed into DCA through deconjugation, dehydroxylation, and dehydrogenation [35]. Colonic epithelial cells covered with biofilms, therefore, are exposed to significantly higher levels of DCA, initiating or aggravating CRC. Further use of the metabolomics analysis is an exciting area of study for investigating direct or indirect metabolic interactions of bacteria and hosts. Taken together, the available evidence strongly suggests that bacterial biofilms are a critical contributor to CRC. Aggregating and invasive pathogens such as *Fn* in biofilms can initiate and progress CRC by eliciting prolonged inflammation and manipulating the host metabolism, which could serve as a biomarker, allowing for earlier drug interventions.

One of the objectives of the human microbiome project has been to identify bacteria and toxins related to CRC. For instance, *Bacteroides fragilis* and *Enterobacteriaceae* bacteria have been found in the mucosal biofilms of patients with inflammatory bowel disease (IBD), increasing host susceptibility to CRC [36]. *Fn* is a prevalent commensal but opportunistic pathogen in human oral cavities and can co-aggregate with almost all types of oral bacteria species associated with oral plaque-forming biofilms [37]. Along with other anaerobic bacteria such as *Campylobacter* and *Leptotrichia*, *Fn* is a key pathogen associated with CRC [38,39] and produces several key virulent factors. One such factor is outer membrane vesicles (OMVs) that contain proteins, lipids, and adhesion molecules. OMVs of *Fn* alone were shown to co-aggregate bacteria and proved to be a powerful biofilm former [40]. OMVs also play a role in bacterial communication and interaction with environments including host cells and immune cells. *Fn* also attaches to and transports non-invasive bacteria into human epithelial cells, functioning as a shuttle vector for other pathogens [41]. *Fn* produces a toxin called *Fusobacterium* adhesin A (FadA) that is correlated with CRC tumorigenesis [13], a lower survival rate [42], metastasis [43], and recurrence [44]. The toxin is responsible for bacterial adhesion, activation of signaling pathways, inflammation, and CRC cell proliferation [45]. Other toxins, such as genotoxins, called polyketide synthase of *Escherichia coli* and enterotoxigenic *Bacteroides fragilis* toxin, have been linked to colon tumors in mouse models [46], but what they cause has yet to be established.

## 3. Biofilm-Associated *Helicobacter Pylori* and Gastric Cancer

*Helicobacter pylori* (*Hp*), classified by the WHO as a class I carcinogen, plays a critical role in the development of gastric cancer, the fifth most prevalent cancer worldwide [47]. Gastric cancer was responsible for ~800,000 cancer-related deaths in 2020 [48]. Chronic *Hp* infection significantly increases the risk of developing gastric cancer and mucosa-associated lymphoid tissue (MALT) lymphoma [49,50]. *Hp* can survive in the low pH of the stomach through its use of urease and pH-sensing BabA [51]. Chronic *Hp* infections create persistent inflammation, provoking the development of gastric cancer. *Hp* also expresses a toxin called cytotoxin-associated antigen A (CagA), which is internalized into host cells via the bacterial type IV secretion system and phosphorylated by kinases such as Fyn and Lyn. It targets and interacts with several factors involved in inflammation and cytoskeleton rearrangement. CagA also activates NF-kB through interactions with a protein called YWHAE (14-3-3ε), influencing cell cycle regulation, signal transduction, and malignant transformation [52,53]. Additionally, CagA has been shown to promote EMT, which contributes to the generation of cancer stem cells [54,55].

The virulent mechanism of *Hp* in the harsh gastric niche appears to be closely linked to its adaptive ability to form biofilms. Various biomedical in vivo systems have demonstrated the existence of *Hp* biofilms in gastric biopsies. Recent transcriptomic and proteomic analyses have provided valuable insights into the transition from planktonic *Hp* to the biofilm phase, including higher gene expression of adhesins, flagella, toxins, efflux pumps, lipopolysaccharides (LPS), the type IV secretion system, urease, and hydrogenase, for obtaining alternative energy sources [56]. Conversely, factors involved in quorum sensing, metabolism, and translation are reduced in biofilms [57]. *Hp* also exhibits adaptive responses during biofilm formation, such as the secretion of OMVs, morphological changes from a spiral to a dormant but more resistant coccoid form, matrix production, and increased efflux pump activity [58]. The altered gene expression of *Hp* is also modulated depending on the stages of biofilm formation, with genes in the initial attachment stage showing different expression patterns compared to the maturation stage of biofilms. Besides the factors mentioned above, there are many other regulators that affect the physiology of *Hp* and promote the transition of planktonic *Hp* to biofilm forms.

Antibiotic treatment of *Hp* is able to reduce the risk of cancer in asymptomatic patients [32] and in individuals with a family history of gastric cancer [59]. It can also reduce the tumor size in gastric MALT lymphoma patients [60]. Biofilm-associated *Hp,* however, is resistant to clarithromycin, a commonly used antibiotic for *Hp* infections. *Hp* biofilms treated in vitro with clarithromycin displayed an increased minimum bactericidal concentration and minimum inhibitory concentration, as well as a higher mutation rate, which contributes to their already significant genotypic variation [61,62]. There is also emerging resistance of *Hp* to other antibiotics such as levofloxacin and metronidazole [63]. In addition to becoming multi-drug-resistant, *Hp* biofilms successfully defend against the host immune response. For example, LPS of *Hp* are recognized via the Toll-like receptor (TLR) 2 system, instead of the common TLR4 system used by most bacteria [64]. The TLR2 pathway mediates weaker host innate and acquired T cell responses in the gastric mucosa, allowing *Hp* to escape effective immune detection [64].

## 4. Tumor Microbiome and Other Cancers

Contrary to a once-believed medical dogma that cancer tissue is sterile, a growing body of evidence has demonstrated the presence of bacteria in various cancerous tumors, including breast, lung, and CRC [27,65,66]. Recent compelling evidence suggests that bacterial communities, known as the tumor microbiome (TM), play a crucial role in tumorigenesis and tumor progression of many different types of cancer. Intriguingly, bacteria have been identified both in the extracellular compartment and inside cancer and immune cells [67]. Immunohistochemistry experiments have demonstrated the presence of Gram-negative bacteria LPS in both the nucleus and cytoplasm of tumor cells, while fluorescence in situ hybridization has detected bacterial 16S rRNA primarily in the cytoplasm. Correlative light and electron microscopy studies have provided visual evidence of bacteria near the nuclear membrane in human breast cancer samples [67]. Moreover, several studies have reported the presence of cell-wall-deficient (L-form) bacteria within breast cancer cells, supporting the notion of their intracellular existence [67,68]. These intracellular bacteria within breast cancer cells can activate Rho-ROCK signaling, reorganize the cytoskeleton, and potentially facilitate metastasis and dissemination to distant parts of the body [68]. Similarly, studies have demonstrated the invasion of *Fn* in esophageal squamous cell carcinoma and its activation of the NF-κB signaling pathway, thereby promoting tumor progression [69]. A toxin such as FadA can also activate β-catenin signaling inside the cancer cells, resulting in increased expression of transcription factors, oncogenes, and inflammatory genes, thereby promoting CRC cell proliferation [19]. The gut microbiota as well as its relation to carcinogenesis and tumor-targeting immunotherapy have been extensively studied. In comparison, the microbiome of other cancers has not been as closely researched. Lungs are a mucosal site extensively exposed to the external environment. Overcoming a technical limitation due to low biomass, the culture-independent 16S rRNA sequencing successfully detected diverse microbes in the lungs [70,71]. Whereas a healthy lung microbiota contains *Prevotella, Streptococcus, Neisseria, Haemophilus*, and *Fusobacterium*, patients with lung cancer and chronic lung diseases such as cystic fibrosis showed dysbiosis [71,72,73]. Higher levels of the *Thermus* genus were correlated with advanced-stage lung cancer [71]. Lower levels of alpha diversity in tumor tissues were detected in lung cancer patients compared to non-malignant lung tissue. Dysbiosis of lung microbiota has also been associated with altered cytokine production and T cell functions, resulting in activation of oncogenic pathways, and promoting inflammation [74]. The pancreas was also once considered to be sterile but has been shown to contain intratumoral microbes [75]. It was found that 76% of the human pancreatic ductal adenocarcinoma harbored bacteria, with *Gammaproteobacteria* being the most abundant taxa. *Proteobacteria* in pancreatic tumor mouse models induced T cell anergy, which suppressed immune responses and promoted tumor progression [76]. In addition to tumors associated with mucosal sites, such as those mentioned above, other types of cancer, including ovarian cancer, bone cancer, and glioblastoma multiforme, have also been shown to associate with the intratumoral microbiome [67,77]. Dysbiosis in the oral microbiome and the presence of TM have also been reported in human head and neck squamous cell carcinoma [78]. However, there is not enough evidence to link specific microbes with these types of cancer.

A recent study utilized a cutting-edge technique called spatial transcriptomics to detect differential gene expression in slices of tumor tissue found in the oral bacterium *Fn* in both CRC and oral squamous tumor cells. This suggests a potential link between oral health and cancers in different parts of the body (Figure 2) [79]. The areas colonized by bacteria exhibited a reduced number of cancer-killing T cells and showed suppressed immune responses. Furthermore, the cancer cells within these immunosuppressive micro-niches displayed significantly lower levels of p53, a tumor-suppressor protein, indicating that the bacteria resided within highly transformed cancer cells in the TME. The oral epithelial cancer cells harboring bacteria showed significant upregulation of genes involved in signaling pathways such as EMT and the interferon-mediated response. The study also revealed that the bacteria-containing epithelial cells were detached from the spheroids and had invaded the surrounding tissue as single cells, thereby promoting cancer progression. In addition, the intratumoral microbiota compromised the effect of a common chemotherapy drug, 5-fluorouracil (5-FU), which also possesses antimicrobial activity [80]. The drug 5-FU was expected to negatively affect *Fusobacterium* in the TME and enhance the effectiveness of anti-cancer therapy. Instead, *E. coli*, a member of the intratumoral microbiota, played a protective role in CRC by metabolizing the drug.

These findings suggest that intratumoral microbes actively contribute to cancer development, progression, and the response to treatment. However, it remains unclear if bacteria simply exploit the pre-established immunosuppressive regions of the TME or if they actively contribute to the immunosuppression within cancer-promoting niches to further thrive. It is also unclear whether a specific intratumoral bacteria or a heterogeneous microbial community such as biofilms is more important in promoting cancer, or if this relationship works differently in different types of cancer. Other reports have demonstrated the presence of fungal species and viruses in the TME as well, suggesting that this effect may not be limited to bacteria [81,82].

While some bacterial species are linked to cancer development, others modulate tumor growth and prevent cancer by exerting anti-tumorigenic effects. *E. coli* from a healthy human gut produced short-chain fatty acids (SCFs), such as acetic acid, which showed cytotoxicity on CRC and breast cancer cell lines and functioned as a probiotic metabolite [83]. Propionate can upregulate tumor necrosis factor alpha-induced protein 1 and promote apoptosis in CRC cells [84].

Lactic acid bacteria can enhance type I interferon production and prevent overactive NF-κB-dependent inflammation in the gut by activating the stimulator of interferon genes and mitochondrial antiviral signaling [85]. The exact mechanisms by which the gut microbiome could function as immunomodulatory agents are still not well understood.

Ultimately, elucidating the intricate interactions between biofilms, the tumor microenvironment, and cancer cells holds great potential for the development of innovative therapeutic strategies that can target these microbial communities and enhance the efficacy of existing cancer treatments. Such knowledge may lead to the discovery of novel biomarkers, the development of personalized treatment regimens, and the advancement of precision medicine in the fight against cancer.

## 5. Therapeutics and Their Clinical Implications

Biofilm-associated cancer therapeutics face a variety of challenges, but there are a variety of emerging therapies that seek to target their joint weakness. One such strategy would be to target an integral bacterium that drives cancer proliferation, which is complicated by the antibiotic resistance offered by the biofilm itself. One promising therapeutic strategy has been to use mucolytic agents such as N-acetyl cysteine (NAC). NAC has been shown to disperse *H. pylori* biofilms in in vivo infections [86], and when used in conjunction with a dual therapy of clarithromycin and a proton pump inhibitor, it has resulted in a significantly greater reduction in *H. pylori* load than the dual therapy alone [87]. These results suggest that NAC may be an effective way to clear carcinogenic infections, but further studies are needed to determine its potential application in antimicrobial therapies.

Other treatment options attempt to simultaneously target biofilms and cancer. N-acyl-homoserine lactones (AHL) are quorum-sensing molecules important for biofilm formation for many bacteria and have been implicated in carcinogenesis for oral squamous cell carcinoma (OSCC) by repressing NF-κB signaling [2,88]. Certain bacteria, such as *Pseudomonas aeruginosa,* use AHL signaling to inhibit bacterial competitors and can even modulate eukaryotic immune cell function [89]. Some synthetic analogs of the AHLs produced by *P. aeruginosa* have demonstrated antimicrobial properties and antiproliferative, or even apoptotic, effects on OSCC cells [89,90]. Biofilms especially seem to provide a protective coating for tumors. Extratumoral *E. coli* biofilms can reduce the efficacy of antibiotics and anti-cancer drugs, create closer cell–cell contacts, improve cancer cell viability, and increase metastatic potential [91]. Treatment with antibiofilm agents, in association with antibiotic and anti-cancer drugs, improved cancer eradication outcomes, demonstrating that the biofilm offered important protection for both the bacteria and the cancer cells [91].

Bacteria are also known to modulate the efficacy and toxicity of existing cancer treatments. For instance, the efficacy of monoclonal antibodies that blockade CTLA-4, such as ipilimumab, in preventing tumor growth are contingent upon T cell interactions with specific *Bacteroides* species [92]. Tumors in germ-free mice models were unresponsive to treatment, but when *B. fragilis* and/or *B. thetaiotaomicron* were reintroduced via oral cultures or fecal material transfers, treatment efficacy was restored [92]. Similar effects are observed with cyclophosphamide, oxaliplatin, and cisplatin, wherein microbiota depletion was associated with a lower anti-tumor efficacy [93,94]. Bacteria can also minimize the collateral damage induced by conventional cancer treatments. For instance, treatment with *Lactobacillus* probiotics after radiotherapy reduces the severity of associated diarrhea [95,96,97].

While bacteria and the biofilms they inhabit can pose significant problems for cancer patients, there is increasing research into ways bacteria can be utilized to treat cancer. One potential benefit of this is that bacteria can be used to selectively target cancer cells, minimizing the general toxicity of conventional treatments. Research in this area is exploring the possibility of using those bacteria as a delivery system for cancer-killing compounds. For instance, *Salmonella typhimurium* are chemoattracted to the microenvironment within the quiescent and necrotic areas of solid tumor models, where they preferentially grow [98]. Attenuated *S. typhimurium* can create large necrotic zones in the center of the tumors and stunt melanoma growth [99]. Other research has used S. *typhimurium* as a vector to deliver prodrug enzymes, immunomodulatory molecules, or toxins that further reduced tumor growth or even induced tumor regression [100]. For example, a thymidine kinase expressed by *S. typhimurium* in conjunction with additional prodrugs allows for targeted treatment of breast and colon carcinomas, resulting in greater reductions in tumor growth than bacteria treatment alone [101]. Modified *S. typhimurium* that expressed interleukin-2 were able to induce tumor regression or slow tumor growth in melanomas and osteosarcomas and prevent the formation of metastases [102,103,104].

By harnessing the immunomodulatory properties of certain bacterial components, researchers aim to enhance the body’s natural defenses against cancer. Bacterial peptidoglycan, LPS, flagella, and nucleic acids are recognized by pattern recognition receptors on macrophages, dendritic cells, and neutrophils, triggering immune responses [105]. The activated immune response includes activation of NF-κB and STAT-3, which enhance anti-tumor activity by stimulating cytokine production from macrophages, dendritic cells, natural killer cells, and B cells, and induce proliferation of immune cells [106,107]. This bacteria-based immunotherapy can be an effective treatment that modulates the host immune system to recognize and target cancer cells with minimal side effects. Bacteria can also be engineered to express specific immune-stimulating molecules to further boost their immunotherapeutic potential. Immune checkpoint inhibitors have shown promising results in enhancing the efficacy of cancer immunotherapies [108]. Immune checkpoint proteins are present on cells, including tumor cells, and are recognized and bound to proteins on immune cells, such as T cells. This interaction prevents a strong immune response from destroying cancer cells as well as healthy cells. Immune checkpoint inhibitors are designed to block checkpoint proteins from binding with their partner proteins, allowing T cells to target cancer cells. Current immune checkpoint inhibitors act against checkpoint proteins, CTLA-4, and programmed death-ligand (PD-1), or its partner protein PD-L1. However, this area of research is still in its early stages, and more investigations are required to optimize the approach and ensure safety in clinical settings.

Another option, one that bypasses the use of biological agents entirely, is to utilize nanoparticles. Nanoparticles typically range in size between 1 and 100 nanometers. Their small size and unique properties can allow them to target specific tissues, including tumor tissues and biofilms, which can offer a more effective and localized delivery of therapeutic agents. The nanoparticles can be designed to penetrate the biofilm matrix to directly deliver drugs to biofilm-infected tumor sites, overcoming one of the major challenges of biofilm treatment—limited drug penetration. For instance, nanoparticles of Paclitaxel, a first-line chemotherapy for gastric cancer, demonstrated greater cytotoxicity at the same dosage compared to Paclitaxel itself, which struggles with low solubility [109]. Moreover, nanoparticles can carry multiple agents simultaneously, allowing for combination therapies that target both the biofilm and cancer cells. For example, calcium fluoride nanoparticles and a graphene/zinc nanocomposite film on artificial dental implants have both demonstrated antibiofilm properties against *Streptococcus mutans*, a known carcinogen [110,111]. This strategy has the potential to enhance the treatment efficacy and reduce the risk of drug resistance, but there are currently several limitations. Organic nanoparticles are often unable to withstand high temperatures or harsh conditions in the body, which makes them poorly suited to address conditions such as gastric cancer [112]. Inorganic compounds, on the other hand, are significantly more stable, and inorganic nanoparticles have been utilized to treat gastric cancer. Zinc oxide nanoparticles (ZnO-NP) have been demonstrated to prevent inhibition, migration, and invasion of gastric cancer cells, and can even induce apoptosis [113]. As with other emerging therapeutic approaches, nanoparticle-based therapies are still in the preclinical stages of research and development, and further research is needed.

The correlation between certain bacteria and corresponding cancers could be used as a biomarker for cancer diagnosis [114,115]. The microbiomes of tumor and healthy tissues are significantly different and vary among tumor stages, tumor grades, and race [116,117,118]. In fact, intratumoral microbiome data are available to distinguish lung, pancreatic, esophageal, and oral cancers in healthy individuals [119,120,121,122]. Long-term survivors of pancreatic adenocarcinoma (PDAC) show high microbial diversity in the TM, suggesting that profiling intratumoral microbiomes could be used to predict PDAC survival in humans [123]. The presence of *Fn* was associated with a shorter survival time for esophageal cancer and showed correlation with its prognosis [124].

Tumor microbiome profiles, while holding great potential as biomarkers, are also difficult to acquire. Factors such as a low microbiome biomass, difficulty in obtaining tumor tissues, ethical issues in obtaining organ biopsies from a healthy human for comparison, and possible contamination of samples all complicate efforts to use the TM as a diagnostic tool. There are alternatives to screening cancers which include the tongue microbiome for pancreatic cancer, the oral microbiome for esophageal cancer, and the blood microbiome for CRC and breast cancer [125,126,127]. Clinical application of the intratumoral microbiome offers a promising avenue for developing cancer screening and more effective treatments to improve cancer patient outcomes.

## 6. Conclusions and Future Directions

Until recently, only dozens of declared bacteria, such as *Fn* and *Hp*, have been considered to cause carcinogenesis by inducing inflammation and altering signal transduction, which influence mucosal cells. Biotechnology utilizing single-cell genomics and multi-omics has provided a new concept: that the TM is present within many tumor tissues and interacts with host cells. The TM is often found in tumor regions where T cell recruitment and its function are suppressed, promoting tumor progression. Considering that the immune-privileged and hypoxic TME can provide an ideal environment where fastidious microbes such as obligate and facultative pathogenic anaerobes bacteria can proliferate and colonize, it is presumed to contain biofilms.

Biofilms are one of the successful forms of life for microorganisms. They provide protection for microorganisms from immune detection and a favorable environment. It is well recognized that biofilms play a key role in cancer progression and that the TME contains many types of microorganisms. However, as research on biofilms and the TM in the TME is still in its early stages, many questions remain unanswered. The composition, dynamics, and functional significance of biofilms and the TM can significantly vary between different cancer types, stages, and individual patients. The unique complexity and adaptation of biofilms may bring about changes in the host’s peripheral environment, metabolism, and immune responses, facilitating host cells’ transformation and tumor progression. While some bacteria and their biofilms mainly function as cancer promoters, growing evidence in clinical settings shows that a healthy gut microbiome and its metabolites can exert anti-tumorigenic effects. Interestingly, it has been suggested that the intestine is the source of intratumoral microbiomes at different locations [128]. Bacterial profiles are similar in both the duodenum and the gut, and a fluorescently tagged bacterial strain has been shown to be translocated from the mouse gut into the pancreas [76]. This indicates that the gut microbiome might translocate from the intestinal tract into other organs via blood circulation and could be feeding various organs, such as the breast, pancreas, lungs, and kidneys, therefore constituting TM at a particular site [67,76,129].

Considering the close relationship between the gut microbiome and the TM in various organs, elucidating the exact mechanisms via which the gut microbiome performs in the human body could improve the therapeutic responses in many cancer patients. Immune modulation using a healthy gut microbiome can be used in cancer immunotherapy. The gut microbiome can be screened before treatment to determine whether the gut contains pathogenic and/or biofilm-forming bacteria. Nevertheless, it is difficult to determine the best microbiome profiles and the function of each bacterial strain in complex environments. Identifying monoclonal bacterial strains functioning as key cancer promoters would be beneficial to anti-tumor therapeutics. Precise characterization of the component in the TM would also provide valuable insight for designing selective drug delivery methods for cancer therapies with less side effects. A comprehensive understanding of biofilm-mediated effects in cancer and the TM in the TME will require further investigation using advanced techniques, such as spatial transcriptomics, single-cell sequencing, and high-resolution imaging. We suggest that future cancer therapeutics should consider combining cancer chemotherapy and bacteriotherapy along with biofilm-disrupting agents. The innovative therapeutic agents can help to more effectively defeat tumor progression, metastasis, and drug resistance.

## Figures and Tables

**Figure 1 ijms-24-12836-f001:**
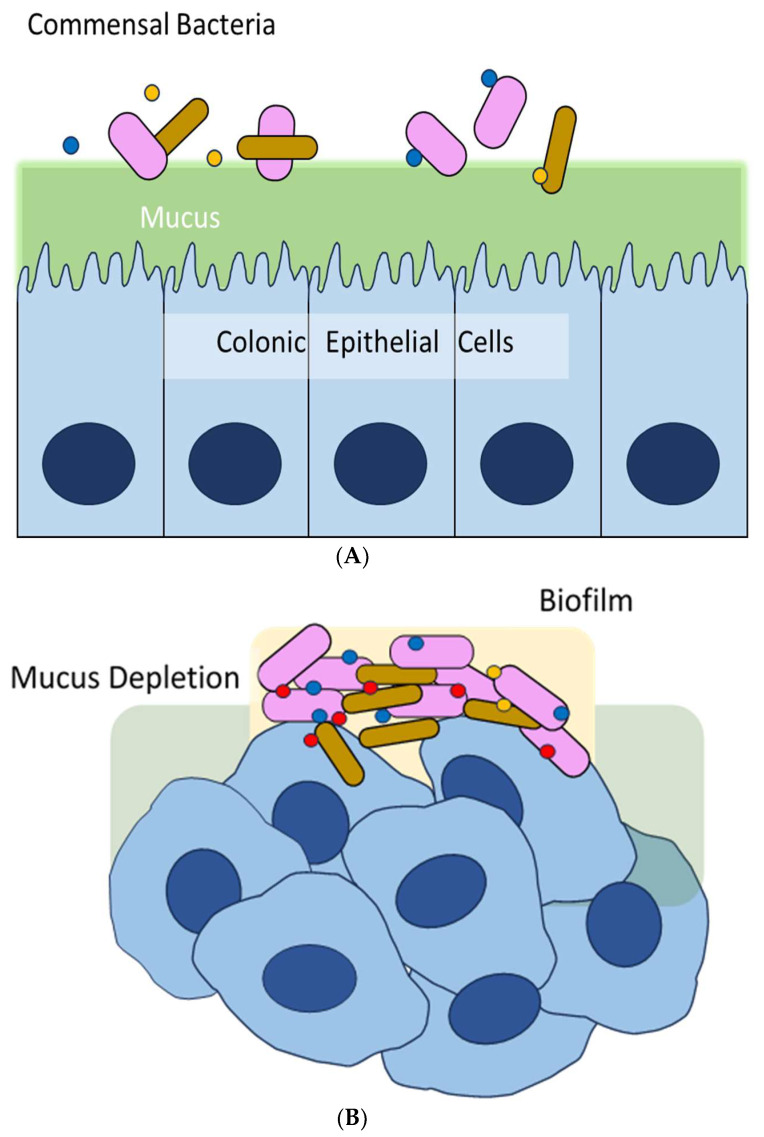
Biofilm-mediated CRC carcinogenesis. (**A**, upper) In healthy individuals, the colon is lined with a dense layer of mucus that separates the microbiota from the human colonic epithelial cells. (**B**, bottom) Biofilms cause mucous depletion and closely contact colonic epithelial cells. This leads to chronic mucosal inflammation and increased levels of IL-6, resulting in the host cell’s transformation and tumor progression.

**Figure 2 ijms-24-12836-f002:**
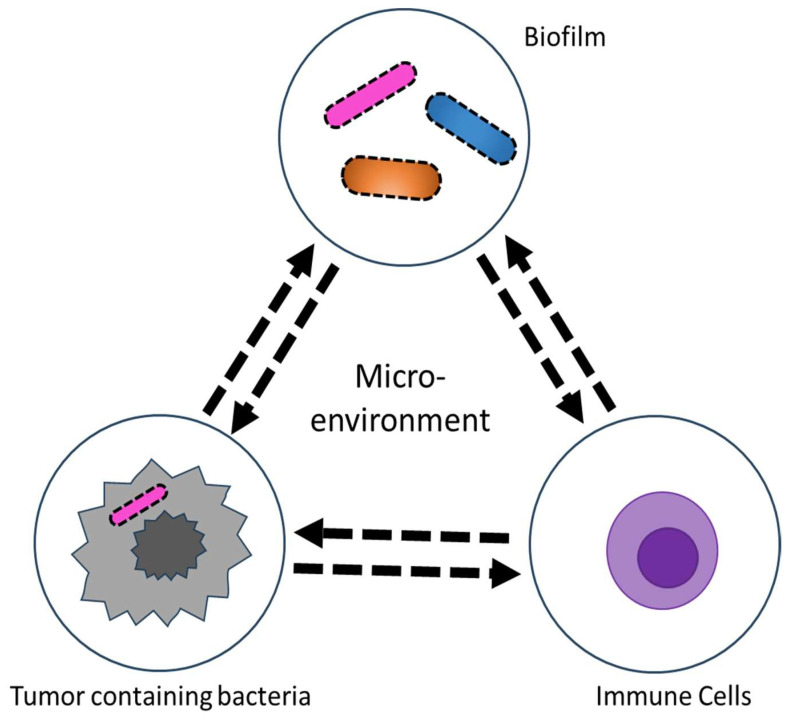
Schematic of the human tumor microenvironment that contains the tumor microbiome. The tumor microbiome provides a unique environment where intratumoral or extra-tumoral microorganisms crosstalk with tumor cells and immune cells.

**Table 1 ijms-24-12836-t001:** Potential mechanisms of biofilms and intratumoral microbes in carcinogenesis: microbial communities in the tumor-containing tissues are associated with cancer, but their exact roles need to be further determined.

Stages of Carcinogenesis	Potential Mechanisms of Biofilm/Intratumoral Microbes	Name of the Microbes and Their Components
Tumor initiation	Promoting mutagenesis and epigenetic alterations by producing substances that cause DNA damage, cell cycle arrest, and instability of genetic materials.	Colibactin produced by some *E. coli* and other Enterobacteriaceae strains causes DNA damage [16,17]. It promotes tumorigenesis in CRC [18].
	Producing or releasing metabolites that induce carcinogenesis in host cells.	Polymicrobial bacteria (see text for details).
Tumor promotion	Activating Wnt/β-catenin signaling pathway, leading to higher gene expression of c-*MYC* and *Cyclin-D-1* (uncontrolled cell division).	Activation of β-catenin by FadA from *Fusobacterium nucleatum* is associated with CRC and gastric cancers [19,20]
	Inducing a chronic pro-inflammatory response.	*Fusobacterium nucleatum* activates NF-κB, a critical factor for cancer, promoting inflammation [13,18].
Tumor progression/Metastases	Direct or indirect immunomodulation by suppressing anti-tumor immune response.	Colibactin-positive *E. coli* is associated with a reduction of CD3+CD8+ T cells in a CRC mouse model [21].
		Clostridial C3 toxins from *Clostridium botulinum* and *Clostridium limosum* disturb phagocytosis and migration of macrophages [22].

	Supporting tumor metastasis.	*E. coli* in the gut stimulates cathepsin K, a metastasis-associated secretory protein, which mediates M2 macrophage polarization in a TLR-4-dependent manner in CRC [23].

## Data Availability

Not applicable.

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
