# Peer review of "Biofilm and Cancer: Interactions and Future Directions for Cancer Therapy"

_ijms, 2023, doi:10.3390/ijms241612836_

Round 1

Reviewer 1 Report

Manuscript ijms-2560250

„Biofilm and Cancer: Interactions and future direction for cancer therapy” for International Journal of Molecular Sciences

Comments:

1.      Please summarize in the form of a table which elements or stages of carcinogenesis are affected by biofilm or its components.

2.      Please also indicate whether the biofilm affects the arachidonic acid pathway and thus the development of inflammation.

3.      Please also indicate whether biofilm as a natural form occurring in the body may have an anti-cancer effect.

Reviewer 2 Report

Dear colleagues.

It has been a pleasure to me to review the present draft.

Despite its high quality I have some general comments:

- Why have you just limited to colon and gastric cancer? What about all the rest of digestive tumors, or even lung and head and neck ones, that obviously are also exposed to such biofilms? This should be clearly stated in the introduction.

- I would also suggest to include a section in the review that will talk about clinical implications in the field.

- An additional section giving "future directions" for research in the field would also be quite interesting.

Round 2
